# Research

ecology, ecosystems

colonization–extinction dynamics, habitat fragmentation, population abundance fluctuations, predation, river connectivity

**Authors for correspondence:**
Carl Tamario
e-mail: carl.tamario@lnu.se
Anders Forsman
e-mail: anders.forsman@lnu.se

# Size, connectivity and edge effects of stream habitats explain spatio-temporal variation in brown trout (*Salmo trutta*) density

Carl Tamario[1], Erik Degerman[2], Daniela Polic[1], Petter Tibblin[1] and Anders Forsman[1]

[1]Ecology and Evolution in Microbial Model Systems, EEMiS, Department of Biology and Environmental Science, Linnaeus University, 392 31 Kalmar, Sweden
[2]Department of Aquatic Resources, Institute of Freshwater Research, Swedish University of Agricultural Sciences, Drottningholm, Sweden

 CT, 0000-0002-3176-130X; ED, 0000-0003-3620-0568; DP, 0000-0001-6168-6489; PT, 0000-0001-6804-5342; AF, 0000-0001-9598-7618

Ecological theory postulates that the size and isolation of habitat patches impact the colonization/extinction dynamics that determine community species richness and population persistence. Given the key role of lotic habitats for life-history completion in rheophilic fish, evaluating how the distribution of swift-flowing habitats affects the abundance and dynamics of subpopulations is essential. Using extensive electrofishing data, we show that merging island biogeography with meta-population theory, where lotic habitats are considered as islands in a lentic matrix, can explain spatio-temporal variation in occurrence and density of brown trout (*Salmo trutta*). Subpopulations in larger and less isolated lotic habitat patches had higher average densities and smaller between-year density fluctuations. Larger lotic habitat patches also had a lower predicted risk of excessive zero-catches, indicative of lower extinction risk. Trout density further increased with distance from the edge of adjacent lentic habitats with predator (*Esox lucius*) presence, suggesting that edge- and matrix-related mortality contributes to the observed patterns. These results can inform the prioritization of sites for habitat restoration, dam removal and reintroduction by highlighting the role of suitable habitat size and connectivity in population abundance and stability for riverine fish populations.

## 1. Introduction

Exploitation resulting in habitat fragmentation, degradation and loss constitute imminent threats to biodiversity worldwide [1–8]. Habitat fragmentation is the process whereby continuous habitats are converted into smaller and more or less isolated habitat islands surrounded by a matrix where environmental conditions are less favourable [2,9,10]. Both the size and spatial arrangement of habitat islands, the quality of the matrix and species characteristics [11] may impact the exchange of individuals between local subpopulations, with important consequences for abundance fluctuations and extinction–recolonization dynamics [12]. This forms the basis of two well-established and related theories in ecology. The theory of meta-population dynamics concerns the roles of inter-patch dispersal for spatio-temporal abundance fluctuations, local extinctions and re-establishments of subdivided populations in fragmented landscapes [12–14]. The equilibrium theory of island biogeography was put forward to explain patterns of variation in species richness of communities on islands, and postulates that the number of species present on an island is determined by the balance between the rates of colonization and extinction, both of which depend on the

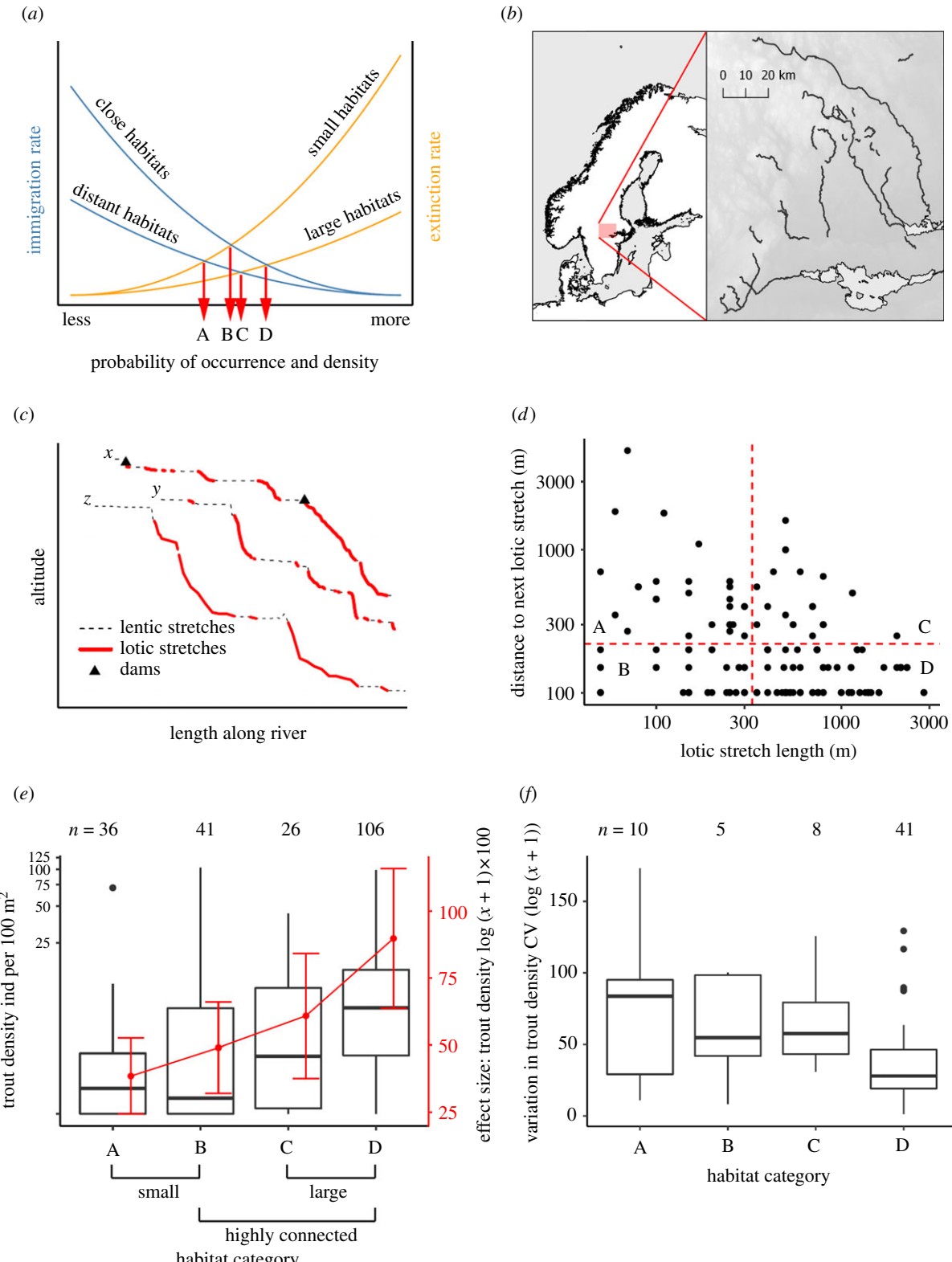

**Figure 1.** Adaptation of the meta-population and island biogeography theory to evaluate whether size and connectivity of lotic river stretches impact density of brown trout. (a) The island biogeography framework used to hypothesize how habitat size and connectivity affect occurrence and density through immigration and extinction processes. (b) The location of the 28 river sections included in this study whose height profiles were extracted. (c) Excerpts of height profiles from three rivers (x, Hammarskogsån; y, Venabäcken; z, Erlandsbobäcken), showing lotic stretches in red lines and lentic stretches in black dashed lines. Distances between lotic stretches were calculated between dams (triangles) to evaluate the role of habitat connectivity for trout population density in free-flowing river segments without constructed barriers. (d) The predicted lotic island habitats were categorized according to size (short and long) and connectivity (close and far), corresponding to categories A–D in a. (e) Brown trout density increased significantly across the four island biogeography categories. Raw data for density including zero-catches are illustrated by boxplots. Predicted means with 95% confidence intervals as acquired from the ZINB mixed model are illustrated in red. (f) The between-year density fluctuations (given as CV—coefficient of variation) decreased with island biogeography category. (Online version in colour.)

size and isolation of the island [15,16] (figure 1a). Although this theory primarily concerns species richness, it spurred the development of the spatially explicit meta-population

framework that recognizes the roles of variation in patch size and isolation [13]. Both theories build on the assumption that smaller islands, or habitat patches, typically harbour

smaller populations (fewer individuals) than larger islands, which increases the risk of extinction owing to demographic stochasticity and reduced genetic diversity [17,18]. Extinction risk in turn may be offset by immigration and recolonization, which is higher in larger and less isolated patches [16].

The island biogeography and meta-population theories were originally developed for terrestrial systems, yet the meta-population dynamic framework has been successfully applied also to river ecosystems where water flow might influence occurrence and abundance variation (e.g. [19–22]). Owing to the topography, geology and hydrology of the landscape, rivers are heterogeneous and contain diverse habitats, ranging from rapid-flowing (lotic) stretches to slow-flowing lake-like (lentic) stretches, each with distinct species assemblages [23,24]. Damming, one of the most widespread environmental alterations of river ecosystems, affects about half of all global large river systems [1,7]. While it is undisputed that damming hinders migration for aquatically bound organisms such as fish [25], it also results in the conversion of lotic stream habitats to lentic habitats, with parallel shifts in fish species assemblages [24,26–28]. Evidence is mounting that rheophilic fragment-dwelling and -spawning species that depend on lotic stream habitats during some life-history stage(s) are declining at alarming rates, in part due to weirs and dams [26–29]. Affected fish species include salmonids of high socioeconomic importance, such as brown trout (*Salmo trutta*), which may impact lower trophic levels, community composition, species interactions (for example, as a host species for the endangered freshwater pearl mussel; *Margaritifera margaritifera*) and ecosystem functioning. A better understanding of what determines the distribution and abundance of riverine fish, and informed projections of how they might respond to management actions and exploitations (e.g. damming, channelling and clearings) that modify river habitats, is therefore in great demand [26].

Reviews and meta-analyses based on data for diverse study systems and organisms (except fish) suggest that there is no consistent relationship between patch size and population density, with positive and negative relationships being almost equally common [30–33]. However, in the largest study conducted so far, based on multiple estimates of the relationship between density and area for 287 species (246 birds, 21 mammals and 20 insects), the mean effect size for individual species showed that there was an overall positive correlation between patch area and population density [34]. This inconsistency emphasizes the importance of considering autecological mechanisms (i.e. how specific species interact with their environment). Migration behaviour, effects of habitat edges and interactions with other species may affect the direction of the predictions [33]. For example, Bender *et al.* [31] report that patch size effects were negative for species that use edge habitats and positive for species that use core habitats.

The extinction–colonization dynamics of meta-populations have traditionally been analysed using spatial occupancy models based on detection–non-detection data [12–14], but richer insights can be gained from abundance-based analyses [35,36]. Empirical evaluations of the patch size–population size paradigm based on multiple-year occupancy data for butterflies, amphibians and birds further suggest that observed local extinctions are more effectively predicted as a direct function of local population size than by patch size [37]. This raises the question how population abundance and dynamics of rheophilic fish is associated with patch size and isolation in river ecosystems.

Here, we analyse data from more than 700 electrofishing events conducted over a 36-year period (1982–2018) in 28 river sections in central Sweden (figure 1*b*). Using a combination of detection–non-detection and population density data, we show that meta-population theory inspired by island biogeography can explain spatio-temporal variation in occupancy and density of brown trout, a riverine fish species present on all continents except Antarctica. Brown trout show exceptional variation in life-history traits [38,39], but the adults generally migrate from natal streams for faster growth and then return to use gravel beds in lotic stretches for reproduction [40]. Lotic habitat is also used by juveniles as nurseries for the first few years. In our study area, even adults seem to reside in lotic habitats outside reproductive events, as these habitats provide oxygen-rich waters, food and refuge from predators, such as pike (*Esox lucius*) that are generally more abundant in or close to the lentic matrix habitats [39,41,42]. The populations included in this study do not express anadromy but individuals may move between lotic stretches or to lakes in search for better foraging opportunities [43] or spawning sites. Inter-habitat dispersal may also allow for recolonization after local extinctions [44,45]. Given the pivotal roles of lotic river stretches and dispersal opportunities for rheophilic fish productivity [39–43,46,47], we hypothesized that the density of brown trout should be positively associated with the size and connectivity of lotic habitats and negatively affected by edge effects associated with elevated predation risk in the vicinity of lentic stretches (figure 1*a*).

## 2. Material and methods

### (a) Habitat identification, verification and classification

Height profiles for 28 river sections in central Sweden were constructed. This was accomplished by combining vector lines of rivers as obtained from the Property Map (fastighetskartan, 1 : 10 000) and altitude data from the national height data model (GSD-Höjddata grid 2+), the highest-resolution layer available of each (acquired from Lantmäteriet; the Swedish mapping, cadastral and land registration authority), respectively, in Sweden. Altitude was extracted every 10 m longitudinally along the river centrelines. To facilitate the calculation of locations, slopes and lengths of lotic habitats from the height profiles, raw data elevation noise was removed with an interpolating formula created through iterative trial and error in Excel (electronic supplementary material, figure S1 and table S1). When a series of interpolated point measurements generated a stretch of at least 50 m with a gradient of at least 0.25%, it was considered a lotic habitat island [48]. A total of 714 lotic stretches were identified with all other stretches in-between considered lentic matrix habitat (figure 1*c*; electronic supplementary material, figure S2).

To confirm the classification of lotic habitats identified by the above GIS method, a field verification was performed in parallel with the GIS work during the summer of 2017. Its objective was to confirm the location of the lotic stretches and to quantify the within-river length of the lotic stretches by the means of a laser-measuring device. Data for 60 verified sites showed that the predicted lengths, as generated by the GIS method, correlated highly to laser-measured lengths (log–log regression, $p < 0.0001$, $F_{1,58} = 317$, adjusted $R^2 = 0.84$). A few (less than 5%) short, lotic stretches predicted through GIS were not encountered in the field. We found these errors acceptable and solely used the spatial data created through GIS in the analyses. That spatial data like this can be created entirely by computer is valuable because of the possibility to extrapolate to larger contexts.

The within-river length of each lotic habitat identified by GIS was used as a measure of habitat (island) size. The within-river distance to the nearest lotic habitat was estimated by GIS both upstream and downstream; the shortest of the two distances was used as a measure of connectivity. See electronic supplementary material, figure S2 for a schematic illustration of the data. Dams impede the free movement of fish, and the national dam registry provided by the Swedish Meteorological and Hydrological Institute (SMHI) was used to locate all dams within the studied area. All calculations of distances to the next habitat island were done within free-flowing segments *between* registered dams. To analyse edge effects, the distance to the lotic–lentic habitat edge (i.e. the distance to the closest lentic habitat) was calculated individually for each electrofishing site (electronic supplementary material, figure S2).

To enable analyses of data within a framework inspired by the island biogeography theory, the lotic habitats were split into four categories (small and large islands with low or high connectivity; figure 1). The delineations were done (lotic stretch length = 330 m and distance to the next lotic stretch = 220 m) to maximize the number of cases in each category (figure 1*d*). A sensitivity analysis (see the section *Sensitivity analysis regarding delineation of the habitat categories*) demonstrates that the use of different cut-off points for size and connectivity did not qualitatively change the overall results and conclusions (electronic supplementary material, figure S3).

## (b) Electrofishing sampling data

Data on the fish presence and density were generated through electrofishing, a non-lethal fish sampling method mainly conducted in streams where it is possible to wade [49]. Electric current (DC) is used to attract fish to swim towards a hand-held anode where they are caught with a dipping net. It is an established and reliable method for quantifying fish density, see Bohlin *et al.* [49] and the Swedish and European Standard [50] for detailed descriptions on the method.

The output from the lotic habitat prediction was overlaid by electrofishing sampling locations from the Swedish Electrofishing RegiSter (SERS at the Swedish University of Agricultural Sciences). On the selected 28 river sections, 343 electrofishing sites were located, which subsequently were filtered to give 209 sites owing to 68 sites being located outside the 714 predicted stream habitats and 66 sites were completely isolated by dams. The 209 electrofishing sites eligible for analyses encompassed 702 electrofishing occasions spanning from 1982 to 2018 and each site was sampled between 1 and 28 times.

## (c) Statistical analyses

Analysing the effects of patch size and isolation using a formal meta-population occupancy- or density-modelling framework [14,35,36] was not possible, because electrofishing sampling data for trout were not available for all lotic river stretches in our study area. Instead, we used the closest predicted neighbouring habitat (and their size), regardless of whether it was occupied or not, as a proxy of connectivity.

### (i) Data handling and model choice

The brown trout density data (expressed as the number of individuals per 100 m$^2$) contained an excess of zeros, resulting in zero-inflation. Although catchability (resulting in false zeros) may play a role in producing zero-catches [49], we do not believe they pose a big problem. False zeros are more likely at very low population densities, so even if there are have some false negatives, these data points will not be especially misrepresentative. In any case, to account for excess of zeros and simultaneously test for local extinctions, we used models with a zero-inflated

**Table 1.** Associations of brown trout density with categorical habitat variables. Results from mixed zero-inflated binomial distribution model on the effects of habitat categories (A, small, far; B, small, close; C, large, far; D, large, close; figure 1) on occurrence and density (count) of brown trout (pooled age classes). The fixed effect coefficient estimates apply to the count data (i.e. density), whereas the zero-part coefficients apply to the probability of encountering zeros (zero-catches). Italics indicate significance at an alpha level of 0.05.

| predictors | estimate | s.e. | Z-value | p-value |
|---|---|---|---|---|
| *count part* | | | | |
| (intercept) | 4.01 | 0.12 | 32.20 | *<0.001* |
| habitat category [B] | 0.46 | 0.15 | 3.10 | *0.002* |
| habitat category [C] | 0.40 | 0.16 | 2.50 | *0.012* |
| habitat category [D] | 0.55 | 0.13 | 4.29 | *<0.001* |
| *zero-inflated part* | | | | |
| (intercept) | −0.85 | 0.60 | −1.40 | 0.161 |
| habitat category [B] | 0.59 | 0.59 | 1.00 | 0.317 |
| habitat category [C] | −0.20 | 0.72 | −0.28 | 0.779 |
| habitat category [D] | −1.97 | 0.71 | −2.79 | *0.005* |

negative binomial (ZINB) error distribution. Trout density was log$_{10}$ +1 transformed, multiplied with 100, and rounded to nearest integer to be compatible with the ZINB model.

As rivers were sampled to different degrees and some lotic stretches were sampled at several locations, there was some dependency between data points. Moreover, some rivers may have higher or lower than average baseline of occurrence and/or density of brown trout due to, for example, differential habitat quality and fishing pressure, possibly obscuring predicted patterns. To account for both of these aspects in the statistical analyses, river identity and stream habitat identity were included as random effects (varying intercept, fixed slope) for both the zero-part and fixed effects, respectively. The ZINB models with mixed effects were performed in R (v. 4.1.0, 18 May 2021) with package glmmTMB (v. 1.1.2).

### (ii) Habitat category classifications

ZINB mixed models were performed to model the effect of habitat category on brown trout density (table 1). River identity and lotic habitat identity were used as random effect in both the count and the zero-part. To test the hypothesis of whether trout density increases linearly with each successive habitat theory category (A–D), we also performed an ANOVA with polynomial contrasts using R, analysing linear, quadratic and cubic relationships. Partial eta squared for the polynomial effects was calculated through equation $SS_{Effect}/(SS_{Effect} + SS_{Error})$.

### (iii) Main model using continuous data

The classifications of habitat size and habitat connectivity into four categories removed a considerable portion of the information contained in these variables; two arithmetically close data points can end up in different categories, which may not be justifiable.

To make better use of the full resolution in the explanatory variables in our dataset, we performed ZINB mixed models in which lotic stretch length (habitat size) and distance to next lotic stretch (habitat isolation) were treated as continuous variables (table 2; model 1). As the size of the nearest neighbouring habitat could influence its ability to act as a

**Table 2.** Associations of brown trout density with continuous variables lotic stretch size, distance to next lotic stretch, directionality (upstream or downstream) and the size of the next lotic stretch, distance to edge of nearest lentic stretch and pike presence/absence. Results from mixed zero-inflated binomial distribution model of the effects of continuous data on size and isolation of lotic river habitats on occurrence and density of brown trout (pooled age classes) (Model 1, AIC = 1837 on d.f. = 14, marginal $R^2$ = 0.125, Akaike weight = 0.9%). The added effects predator and edge effects (Model 2, AIC = 1828 on d.f. = 17, marginal $R^2$ = 0.176, Akaike weight = 99.1%). The fixed effect coefficient estimates apply to the count data (i.e. density), whereas the zero-part coefficients apply to the probability of encountering excess zeros (zero-catches). Italics indicate significance at an alpha level of 0.05.

| predictors | Model 1: only spatial variables | | | | Model 2: added predator and edge effects | | | |
|---|---|---|---|---|---|---|---|---|
| | estimate | s.e. | Z-value | *p*-value | estimate | s.e. | Z-value | *p*-value |
| *count part* | | | | | | | | |
| (intercept) | 5.47 | 0.73 | 7.45 | *<0.001* | 5.76 | 0.75 | 7.67 | *<0.001* |
| distance to next habitat | −0.29 | 0.10 | −2.81 | *0.005* | −0.31 | 0.10 | −3.05 | *0.002* |
| upstream | −0.78 | 0.69 | −1.13 | 0.258 | −1.22 | 0.70 | −1.74 | 0.082 |
| size of habitat | 0.11 | 0.05 | 2.13 | *0.033* | 0.10 | 0.06 | 1.71 | 0.088 |
| size of neighbouring habitat | −0.05 | 0.04 | −1.18 | 0.237 | −0.02 | 0.04 | −0.54 | 0.589 |
| distance to next habitat × directionality (U) | 0.18 | 0.13 | 1.38 | 0.169 | 0.26 | 0.13 | 1.97 | *0.048* |
| distance to edge | — | — | — | — | −0.05 | 0.03 | −1.38 | 0.167 |
| pike presence (1) | — | — | — | — | −0.83 | 0.24 | −3.37 | *0.001* |
| distance to edge × pike presence (1) | — | — | — | — | 0.13 | 0.05 | 2.59 | *0.010* |
| zero-inflated part | | | | | | | | |
| (intercept) | 3.97 | 2.83 | 1.40 | 0.161 | 3.97 | 2.83 | 1.40 | 0.161 |
| size of habitat | −0.94 | 0.31 | −3.05 | *0.002* | −0.94 | 0.31 | −3.05 | *0.002* |
| distance to next habitat | 0.01 | 0.34 | 0.03 | 0.979 | 0.01 | 0.34 | 0.03 | 0.978 |

source of immigrants, we also included the size of the next lotic stretch. Moreover, we included the interaction between the distance to the next lotic stretch and the direction (upstream versus downstream) as matrix penetration might be asymmetric due to flow direction and migration behaviour of brown trout. To analyse predator-mediated edge effects, we added pike presence, distance to habitat edge from the sampled site and their interaction to the above model (table 2; model 2). The parsimony of these two models were compared using Akaike information criterion (AIC). River identity and lotic habitat identity were used as random effects in both the count and the zero-part.

### (iv) Between-year density fluctuations

To evaluate whether and how the size and isolation of lotic stretches influenced temporal density fluctuations of brown trout, the coefficient of variation (CV) in density was calculated using data for the subset of electrofishing sites that had been sampled on at least three occasions ($n = 64$ sites). CV in brown trout density on an electrofishing site was not associated with number of electrofishing occasions (linear regression, $F_{1,62} = 0.48$, $p = 0.49$; electronic supplementary material, figure S4), but was negatively associated with mean density ($F_{1,62} = 54.06$, $p < 0.001$; electronic supplementary material, figure S4). An ANOVA with polynomial contrasts was performed to analyse linear or nonlinear trends over the four habitat categories. Partial $\eta^2$ for the polynomial effect was calculated through the equation $SS_{Effect}/(SS_{Effect} + SS_{Error})$. No random effects were applied.

### (v) Sensitivity analysis regarding delineation of the habitat categories

A sensitivity analysis was performed to evaluate whether the cut-off points used to delineate the two continuous lengths and isolation of lotic stretches variables into categorical variables

(short or long and near and far), influenced results and conclusions. To this end, model outcomes were compared using AIC for different combinations of cut-off points for lotic stretch length and isolation. The delineations seen in figure 1d were set at each 10th percentile (0, 10, 20, …, 100%) for both variables to evaluate which delineations would produce the most parsimonious model. The habitat category ZINB model was performed with all (11 × 11 =) 121 possible combinations of delineations, and the AIC for each model was plotted in a matrix to find the most parsimonious delineation(s) (electronic supplementary material, figure S3).

### (vi) Age classes

The quality control at SERS classifies the caught brown trout into young-of-the-year (0+ or fry, i.e. an estimate of recruitment) and parr and older taken together. Since habitat characteristics and migratory processes may affect these age classes differently, the models were performed using the age classes separately (0+, fry; and 1+, parr and older) and pooled together. Because the results of the separate analyses of the two age classes were qualitatively similar (they never responded qualitatively differently), we report the results from pooled data in the main results. The results from the analyses split by age class are reported in the electronic supplementary material.

## 3. Results and discussion

### (a) Changes in population density and local extinction rates with habitat size and connectivity

In the first approach, lotic stretches were categorized according to their size (long or short) and isolation (close to or far from nearest lotic habitat; figure 1a,d). The results provide

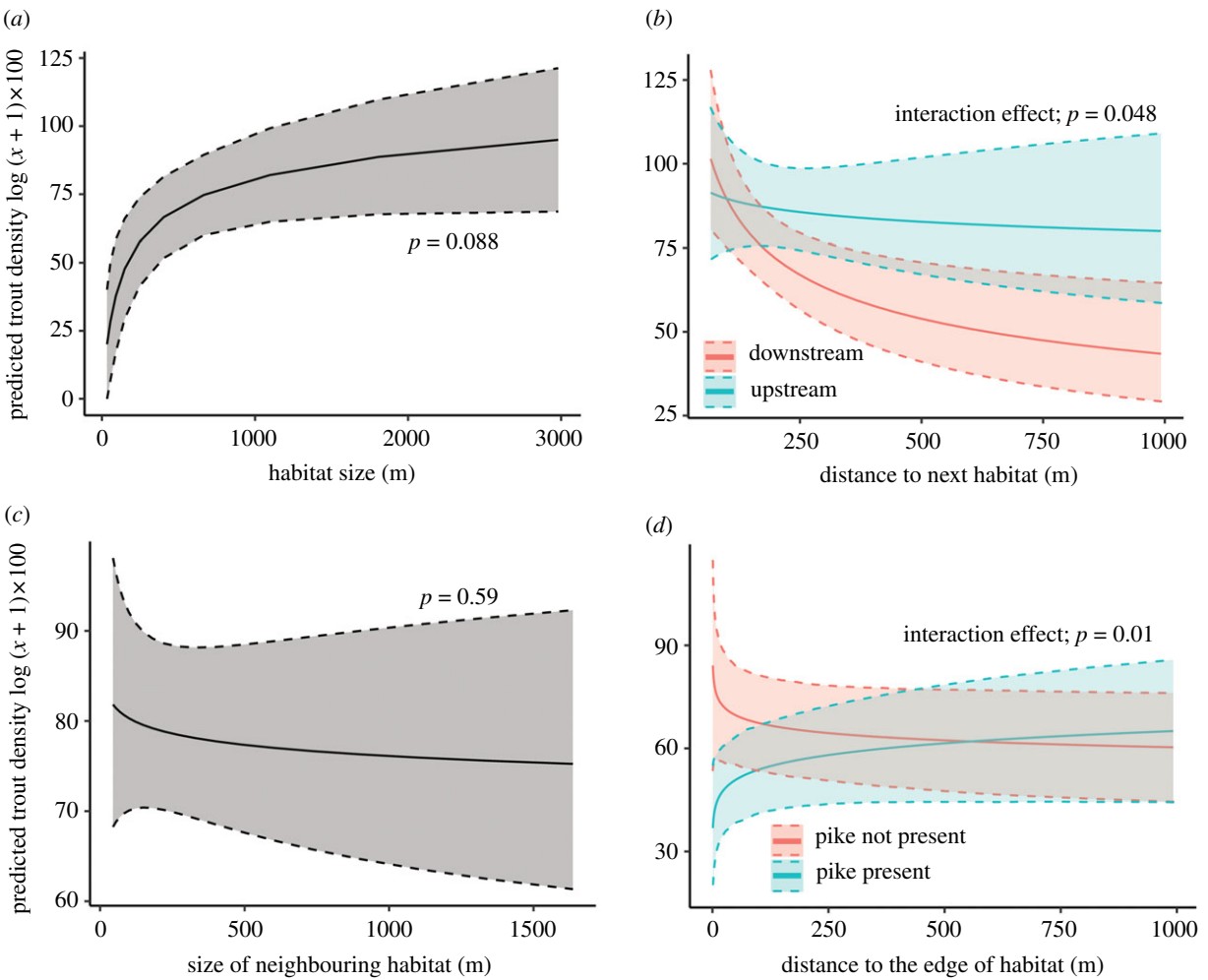

**Figure 2.** The results of ZINB model on brown trout density treating variables of lotic stretch length and connectivity as continuous variables. Effect plot (of predicted mean with 95% CI) illustrating the effects on brown trout density of (*a*) island size, (*b*) the significant interaction effect between 'distance to next lotic habitat' and 'directionality' (whether the next lotic stretch was located upstream or downstream) showing that the positive effect on density of the neighbouring habitat decreases faster with the distance downstream than upstream and (*c*) the size of the next island (for statistical results, see table 2). (*d*) The edge and predatory effects show that trout density decreased with proximity to habitat edge, provided pike had been caught on at least one sampling occasion in the habitat, indicating a strong negative predator-mediated edge effect. (Online version in colour.)

evidence that brown trout density increases progressively with increasing size and decreasing isolation of lotic stretches (ANOVA, linear contrasts, $F_{1,205} = 39.93$, $p < 0.001$, partial $\eta = 0.163$, quadratic contrasts, $F_{1,205} = 1.51$, $p = 0.22$, cubic contrasts, $F_{1,205} = 0.07$, $p = 0.79$; figure 1*e*; see electronic supplementary material, figure S5 for age classes separately). The results from a ZINB mixed model, taking into account both non-zero and zero-catches, confirm the positive trend and further show that large and highly connected habitats not only have higher density of trout but also lower predicted risk of samples with zero-catch, indicative of local extinctions (figure 1*e* and table 1; see electronic supplementary material, table S2 and figure S6 for age classes separately). The delineation of habitats into four habitat categories was made to yield an approximately equal number of cases and spread in each category (figure 1*d*). However, a sensitivity analysis showed that results are representative and not qualitatively influenced by the delineation of the four categories (see Material and methods, and electronic supplementary material, figure S3). Reassuringly, the findings reported above were robust irrespective of the analytical approach. The conclusion that the probability of trout occurrence was higher in longer lotic stretches, and trout density was higher in longer and more connected stretches, remains when size and isolation

were treated as continuous predictor variables (figure 2*a* and table 2; Model 1; see electronic supplementary material, tables S3 and S4 for age classes separately).

The finding that the probability of occurrence and average density of trout populations increase with size and connectivity of lotic stretches is in accordance with the expectations from meta-population theory [12–14], shares a resemblance to the island biogeography paradigm for species richness [15,16] and is in overall agreement with previous studies on occurrence of riverine fish. For example, the probability of the occurrence of bull trout, *Salvelinus confluentus*, in streams within the Boise River basin of central Idaho, USA, was significantly related to stream catchment area and isolation [51]. The probability of Chinook salmon, *Oncorhynchus tshauytsha*, nest occurrence across a stream network in Idaho increased with habitat size, and connectivity was more important for small than for large patches [20]. The probability of the occurrence of stream-dwelling Dolly Varden charr, *Salvelinus malma*, across 78 tributaries in the Sorachi River basin, Hokkaido, Japan, increased with tributary size while being largely independent of isolation [22]. Our present study of brown trout thus adds an important dimension to the existing knowledge contributed by these earlier studies, in that our results provide rare evidence that it is not only the occurrence but also the

density of rheophilic fish populations that increase with size and connectivity of lotic stretches.

The underlying mechanisms responsible for the positive association of trout density with size, and connectivity, of lotic habitats reported here are not clear, but the effects are similar for both age classes. However, if larger habitats are on average richer in resources (e.g. food, microhabitat variability, suitable spawning substrates), this may increase settling rates of migrants, reduce emigration, and promote survival, growth and reproductive output, altogether resulting in higher density. Larger habitats also have higher area-surface/edge ratio than smaller habitats and are therefore, less influenced by matrix-associated edge effects, such as matrix-dwelling predators [11,52].

## (b) Extending with edge effects and the role of predation

Brown trout is susceptible to predation in lentic (matrix) habitats [39,41,42]. Matrix-dwelling predators, such as pike, a keystone piscivore, may enter lotic habitats and cause a negative edge effect through predation or displacement of trout. Consistent with this hypothesis, we found that trout density in lotic stretches increased with distance to the nearest lentic habitat, provided pike was detected on at least one sampling occasion (figure 2d and table 2; Model 2; see electronic supplementary material, tables S3 and S4 for age classes separately). The addition of distance to habitat edge, pike presence and their interaction to the previous model increased model performance significantly (Model 2, AIC = 1828 versus Model 1, AIC = 1837), suggesting that predator presence can partly explain the observed patterns. However, we cannot ascertain based on the available evidence whether the negative impact on trout density indicated by this edge effect reflects an increased mortality of trout due to higher predation rates closer to lentic habitats, or whether it results instead from a behavioural shift in microhabitat use mediated by avoidance of perceived predation risk, or a combination of the two.

The positive effect of connectivity on trout density (figure 1e, tables 1 and 2) probably also reflects in part lower matrix-related mortality of immigrants, since long stretches of lentic habitat may constitute partial barriers for migration through increased predation [53,54]. Further, we only evaluated the connectivity effects mediated by the closest lotic habitat within the main channel without considering the potential habitats beyond the closest habitat and the true dendritic structure of rivers [6]. The results may therefore underestimate the importance of connectivity, because effects of potential recruitment and immigration from tributaries and more distant habitats might also impact density in the analysed lotic habitats.

This latter analysis further unveiled the role of connectivity and inter-habitat dispersal in rivers, because the positive effect on trout density of immigration from neighbouring habitats decreased faster with distance to downstream than to upstream lotic stretches (as evidenced by a significant interaction between direction and distance to nearest neighbouring lotic habitat, $p = 0.048$, figure 2b and table 2). This shows that penetrating the matrix is direction-biased, and probably reflects that it is more difficult for fish to move upstream against the flow than downstream with the flow [55]. Trout density was not associated with the size of the nearest lotic habitat in any of the models (table 2 and figure 2c), pointing to the conclusion

that even small subpopulations may provide immigrants that contribute to increased productivity [56]. In addition, the closest habitats, despite being small, may constitute important stepping stones that provide temporary refuge for individuals migrating from further away [57].

## (c) Large and connected habitats stabilize density fluctuations

Besides explaining spatial variation in the probability of occurrence and average density of brown trout populations, the results show that the size and isolation of lotic habitats modulate temporal density fluctuations (figure 1f; see electronic supplementary material, figure S7 for age classes separately). The between-year variability in population density, estimated as CV in density at each location for the subset of electrofishing locations that were sampled in at least three different years ($n = 64$), decreased progressively across the four habitat categories (ANOVA, linear contrasts, $F_{1,60} = 11.93$, $p = 0.001$, quadratic contrasts, $F_{1,60} = 0.61$, $p = 0.43$, cubic contrasts, $F_{1,60} = 0.75$, $p = 0.38$; figure 1f). That populations in longer lotic stretches appear to be more stable might reflect that they were on average more abundant (electronic supplementary material, figure S4) and therefore less influenced by environmental and demographic stochasticity [18], and presumably also less likely to go extinct [9], although sample size-related variability might also have contributed to this pattern [49]. The finding that less isolated populations were more stable is consistent with a dampening effect of dispersal on density fluctuations [31].

A recent study based on 18 years of observations of four ecologically distinct species of fish in Japan indicates that tributary branching complexity can stabilize watershed meta-populations of lotic and semi-lentic fish [21]. Theoretical simulations and time-series data of fish populations representing a large number of species across Europe further indicate that network connectivity and branching complexity can buffer fish meta-populations against synchronous dynamics [19]. Together, these studies suggest that variance reducing portfolio effects associated with variation among populations across environments may increase stability, productivity and resilience of species and ecosystems [58–60]. However, our present results provide the first demonstration, to our knowledge, that the large size and high connectivity of lotic stretches can have a stabilizing effect on individual populations of riverine fish. A remaining unknown is how the individual-population-level stabilizing effect balances with the potential synchronizing effect of increased dispersal to determine the overall effect on meta-population-level variability.

## 4. Conclusion

A key finding emerging from the present study is that spatially derived high-resolution habitat data as estimated from altitude and river polylines alone can explain a substantial amount (marginal $R^2 = 0.12$–0.17) of the spatial density patterns and temporal density fluctuations of fish populations in rivers. The analyses provide evidence suggesting that large and well-connected rapid-flowing riverine habitats positively promote habitat occupancy, increase and stabilize population density, and reduce local extinction rates of brown trout—S. trutta. The results further suggest that the density of trout in lotic habitats can be

suppressed by edge effects mediated by predation from pike. As such, this work advances the understanding of how physical, geometrical and biotic characteristics of fragmented river habitats together influence the spatial distribution, density and dynamics of fish populations.

There is some evidence from previous studies based on occupancy data that the size, connectivity and branching complexity of river networks may impact the occurrence, stability and synchrony of meta-populations of riverine fish species [19–22]. Our present finding that trout density—but not the probability of excess zeros—was significantly associated with habitat connectivity supports the conclusion that analysing meta-populations based on density data provides higher resolution than with occupancy data and can help detect ecologically important processes [35,36].

An increased focus on how the properties of free-flowing river sections between dams affect productivity and resilience may improve the success of habitat restoration and conservation programme aiming to revitalize declining fish populations. Based on the results here, prioritizing measures that enable previously isolated stretches to be connected will be most effective, and the larger the stretches the higher the expected density and stability of populations. Dam removals, for example, could restore connection but also create new lotic habitats from previously inundated areas, generating both productivity in the restored lotic stretch *and* new migration stepping stones for rheophilic fish. On a more general note, recognizing the roles of size and distribution of lotic habitats may inform environmental engineering, improve the sustainability of hydroelectric energy production and ultimately aid the protection of biodiversity and ecosystem functioning. This is key in an era of accelerating exploitation and river fragmentation.

In view of the growing threats to biodiversity [2–4,26–28], our findings are encouraging because they imply that management actions aimed at dam removal and habitat restoration not only provide positive effects mediated by colonization and utilization of previously blocked-off areas, but also suggest additive positive effects by increased habitat connectivity on viability of local populations of brown trout.

Data accessibility. The datasets generated and analysed during the current study are available from the Dryad Digital Repository: https://doi.org/10.5061/dryad.7pvmcvdt5 [61]. The data are provided in electronic supplementary material [62].

Authors' contributions. C.T.: conceptualization, data curation, formal analysis, investigation, methodology, project administration, resources, software, validation, visualization, writing—original draft, writing—review and editing; E.D.: conceptualization, investigation, methodology, supervision, validation, writing—review and editing; D.P.: conceptualization, investigation, methodology, validation, writing—review and editing; P.T.: conceptualization, funding acquisition, investigation, methodology, project administration, supervision, validation, writing—review and editing; A.F.: conceptualization, formal analysis, funding acquisition, investigation, methodology, project administration, supervision, validation, writing—original draft, writing—review and editing. All authors gave final approval for publication and agreed to be held accountable for the work performed therein.

Competing interests. The authors declare no competing interests regarding funding, current or previous employment(s) or personal financial interests.

Funding. This work was financially supported by The Swedish Research Council Formas (Dnr. 2017-00346 to A.F. and P.T. and Dnr. 2018-00605 to P.T.), Stiftelsen Oscar och Lili Lamms Minne (DO2017-0050, to A.F. and P.T.) and Linnaeus University (to A.F. and P.T.).

Acknowledgements. We thank M. Dopson and two anonymous reviewers for comments on the manuscript and language.

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
