## [Peer Review File · Proceedings of the Royal Society B: Biological Sciences]

Review History

RSPB-2021-1255.R0 (Original submission)

Review form: Reviewer 1

Recommendation

Accept with minor revision (please list in comments)

Scientific importance: Is the manuscript an original and important contribution to its field?

Good

General interest: Is the paper of sufficient general interest?

Good

Quality of the paper: Is the overall quality of the paper suitable?

Good

Is the length of the paper justified?

Yes

Should the paper be seen by a specialist statistical reviewer?

Yes

Do you have any concerns about statistical analyses in this paper? If so, please specify them explicitly in your report.

No

It is a condition of publication that authors make their supporting data, code and materials available - either as supplementary material or hosted in an external repository. Please rate, if applicable, the supporting data on the following criteria.

Is it accessible?

Yes

Is it clear?

Yes

Is it adequate?

Yes

Do you have any ethical concerns with this paper?

No

Comments to the Author

This paper provides an analysis to show that patch size and distance correlate with population density. This is a fairly fundamental question in spatial ecology, well-explored in a variety of taxa as the authors reference, while this paper investigates it in the context of riverine fish in particular. My first main feedback is one of this framing: currently, the introductory text on the relationship between patch size and population sizes in other taxa (lines 93-99) might be taken to imply that there is no support for such a relationship. It is not until the discussion (lines 291-298) that the authors clarify that half of studies in the referenced meta-analyses do support such a relationship, and a common context-dependency for why mixed results might occur is whether or not a species uses core or edge habitat (with a patch size/abundance relationship more likely to occur for core habitat species). This paper would be much better served by clarifying this context from the beginning, in the introduction, and then motivating why this relationship might require a new analysis for the case of riverine fish (e.g., could advection change these expectations?).

Second, the description of the potential applications of the findings here to biodiversity conservation (abstract lines 43-46, discussion lines 385-396) could use greater specificity: how exactly might one use these findings to prioritize sites for habitat restoration? Also, wouldn't dam removal (line 393) lead to more lotic "matrix" habitats rather than lentic "patch" habitats, such that there'd be a trade-off with dam removal between positive effects on connectivity and negative effects on patch number and size for lentic species (as indicated on lines 80-81)?

Third, for the results on population variability, to claim support for the portfolio effect specifically (lines 350-353), the authors must demonstrate that the coefficient of variation would be greater if all sub-populations were a single homogenous population instead of divided into sub-populations (see the analysis in the Schindler et al 2010 paper cited here). The portfolio effect means not just lower variability, but specifically lower variability of a meta-population arising from independence of sub-populations (e.g., each sub-population having independent sub-population-level fluctuations due to factors such as localized environmental variability or local adaptation, so a bad year for one sub-population isn't necessarily a bad year for all sub-populations). In addition, while larger population sizes in sub-populations can increase portfolio-level buffering, anything that increases connectivity can, in some cases, reduce portfolio buffering by increasing cross-sub-population synchrony. See:

Abbott, K. C. (2011). A dispersal-induced paradox: synchrony and stability in stochastic metapopulations. *Ecology Letters*, 14(11), 1158-1169.

Bjørnstad, O. N., Ims, R. A., & Lambin, X. (1999). Spatial population dynamics: analyzing patterns

and processes of population synchrony. *Trends in Ecology & Evolution*, 14(11), 427-432.
 Liebhold, A., Koenig, W. D., & Bjørnstad, O. N. (2004). Spatial synchrony in population dynamics. *Annu. Rev. Ecol. Evol. Syst.*, 35, 467-490.

for more on this. If, instead of doing a portfolio effect analysis to support their claim here in the discussion, the authors want to connect their results to the portfolio effect concept without additional analyses, then they would need to recognize its multiple facets (e.g., the potential role of connectivity in synchronizing populations) in a more careful and nuanced discussion and be careful to avoid wording that suggests support for the portfolio effects from their results.

Additional comments:

Line 89: does exploitation modify habitat as implied here? In marine systems, that depends on the fishing gear (with bottom trawling and dynamite fishing in particular causing habitat damage), but I'm not aware of how this occurs in freshwater habitats.

Line 97: why is this a "sad" outcome? Delete "sadly" or use a less normative term (although I expect this sentence will change more substantively in response to the comments above).

Line 143 "that we regarded highly": this phrasing is a bit awkward, maybe "The fact that spatial data like this can be created entirely by computer is valuable because..."?

Line 179-180 "the next by": not sure what this phrase is referring to, perhaps delete?

Line 194: I'm guess there were no data on habitat quality and fishing pressure, but if there were, that'd be quite useful to include in the analyses.

Line 280: for improved clarity here, specify "previous studies" as "occurrence studies of riverine fish", and then replace "As such" with "For example"

Line 315: is a 0.7% difference in AIC really "significant"? It's an improved fit, but seems like a small one.

Lines 369-372: please indicate percent of variation explained to back this up

Lines 373-4 "promote/increase/stabilize/reduce": this phrasing suggests causality was demonstrated, while the data here are observational; either change to phrasing like "are associated with" or soften with "likely" or "might".

Review form: Reviewer 2

Recommendation

Major revision is needed (please make suggestions in comments)

Scientific importance: Is the manuscript an original and important contribution to its field?

Excellent

General interest: Is the paper of sufficient general interest?

Good

Quality of the paper: Is the overall quality of the paper suitable?

Good

Is the length of the paper justified?

Yes

Should the paper be seen by a specialist statistical reviewer?

No

Do you have any concerns about statistical analyses in this paper? If so, please specify them explicitly in your report.

Yes

It is a condition of publication that authors make their supporting data, code and materials available - either as supplementary material or hosted in an external repository. Please rate, if applicable, the supporting data on the following criteria.

Is it accessible?

Yes

Is it clear?

No

Is it adequate?

Yes

Do you have any ethical concerns with this paper?

No

Comments to the Author

In this study, authors exploited a large Swedish electrofishing database to assess the roles of lotic habitat size and isolation in shaping the occurrence, density and stability of brown trout subpopulations.

This is a very interesting and robust study, with striking results that match theoretical expectations surprisingly well (given the use of empirical data). It is rather well written and nice to read. I have two main comments though.

My first comment is about statistical analyses. I think that models should have been run with two (nested) random effects, while the authors only considered a single random effect. This would probably not alter the results and interpretations (that seem pretty robust), but this is to be checked.

My second comment is about the discussion, which gets poorer toward the end. The two last sections seem rather incongruous. The former (about between age classes comparisons) could be disseminated throughout other sections, while the latter (about so-called large-scale patterns) is not supported by any clearly detailed hypotheses or statistical analysis and thus appears highly speculative (if not flawed). I suggest that the authors simply delete this last section, or provide full details.

See detailed comments below.

SUMMARY

No specific comment

INTRODUCTION

L.100-103: "autecological mechanisms": This term, as well as related mentioned mechanisms (immigration behaviours and edge effects) are detailed in Bowman et al. 2002 (cited reference) but would here require additional information for readers to fully grasp your statement. This is all the more important that these mechanisms are central in your study.

MATERIAL AND METHODS

L.126: "fastighetskartan": this must be a Swedish word for some kind of map, but please, provide a English translation.

L.134: “with all stretches”: with all other stretches?

L.183: I think that this section is more about “model choice” or “model specification” than about “model selection” (the latter referring to the specific statistical approach of selecting the most parsimonious model among a set of candidate models)

L.195-196: Your design is actually nested: several data points within each lotic stretch, and several stretches within each river. With a single random effect (lotic stretch), you take into consideration the “dependency between data points” but you do not take into consideration the fact that stretches from a same river might provide higher/lower yields. I think that you need two nested random effects: a “stretch” random effect, nested into a “river” random effect. This might not change your results drastically, but I think it would be preferable... all the more so as you actually identified an effect of rivers on densities (Table 3... but see my comment about L.360-366)

L.203 and L.227: ANOVAs with non-linear trends are not reported in the manuscript (unless I am mistaken, only results of ANOVAs with linear trends are reported, in L. 254 and L. 336). Is there a reason for that? Were there inconclusive? Furthermore, could you provide the R-package used for these analyses?

L.229: “No random effects were applied.”: could you explain why? as for densities, I guess that CV could be influenced by both site- and river-peculiarities.

RESULTS & DISCUSSION

L.297: Bender et al.

L.315: please, specify which model (1 or 2) is associated with each AIC

L.320: or even a combination of both? (e.g. 0+ mortality and 1+ behavioral avoidance)

L.355-358: this section seems a bit disconnected from the rest of the discussion and could be merged with the corresponding sections...

L.360-366: this section, as the previous one, is rather “out of the blue”, and quite obscure, since the corresponding analytical approach is not provided in the method section (unless I missed something). I cannot figure out what you actually did here... Was it a ZINB mixed model? A subset of Model 1?... From my understanding, eta-squared are computed when the model includes categorical predictors, but I think that stretch length and connectivity are continuous predictors here... are you sure of the use of eta-squared? Apparently, you used rivers as a random effect, since discarding the river effect makes you switch from a GLMM to a GLM (caption in Table 3; see by the way my comment about L.195), but at the same time, you mention in L.364 that you used a “random effect of habitat identity” (and not “river identity”): this is to be clarified. Finally, why did you run the model with and without the random effect, and how can you draw any (or such) biological conclusions thereby? All of this has to be clarified... or the whole section just discarded, since the remaining of the manuscript is quite convincing and this section does not add any relevant/robust interpretations.

CONCLUSION

No specific comment

FIGURES AND TABLES

Table 1: “near” versus “close” : you should use a single term

Figure 2 and Table 2: the former uses the term “directionality” (which makes sense and is also used in the main text in L.215) whereas the latter uses the term “upstream”. You should use “directionality” throughout the text (with for instance “Directionality (U)” in Table 2)

Table 3: the statistical approach corresponding to these outputs cannot be found anywhere (or at least, not in a straightforward way). See my comments about L. 360-366.

Decision letter (RSPB-2021-1255.R0)

05-Aug-2021

Dear Mr Tamario:

Your manuscript has now been peer reviewed and the reviews have been assessed by an Associate Editor. The reviewers' comments (not including confidential comments to the Editor) and the comments from the Associate Editor are included at the end of this email for your reference. As you will see, the reviewers and the Editors have raised some concerns with your manuscript and we would like to invite you to revise your manuscript to address them.

Research ethics:

Use of animals and field studies:

It is a condition of publication that you make available the data and research materials supporting the results in the article. Please see our Data Sharing Policies (<https://royalsociety.org/journals/authors/author-guidelines/#data>). Datasets should be deposited in an appropriate publicly available repository and details of the associated accession number, link or DOI to the datasets must be included in the Data Accessibility section of the

article (<https://royalsociety.org/journals/ethics-policies/data-sharing-mining/>). Reference(s) to datasets should also be included in the reference list of the article with DOIs (where available).

[http://datadryad.org/submit?journalID=RSPB&manu=\(Document not available\)](http://datadryad.org/submit?journalID=RSPB&manu=(Document%20not%20available)), which will take you to your unique entry in the Dryad repository.

Please submit a copy of your revised paper within three weeks. If we do not hear from you within this time your manuscript will be rejected. If you are unable to meet this deadline please let us know as soon as possible, as we may be able to grant a short extension.

Best wishes,
Dr Daniel Costa
mailto: proceedingsb@royalsociety.org

Reviewer(s)' Comments to Author:

Referee: 1

Comments to the Author(s)

This paper provides an analysis to show that patch size and distance correlate with population density. This is a fairly fundamental question in spatial ecology, well-explored in a variety of taxa as the authors reference, while this paper investigates it in the context of riverine fish in particular. My first main feedback is one of this framing: currently, the introductory text on the relationship between patch size and population sizes in other taxa (lines 93-99) might be taken to imply that there is no support for such a relationship. It is not until the discussion (lines 291-298) that the authors clarify that half of studies in the referenced meta-analyses do support such a relationship, and a common context-dependency for why mixed results might occur is whether or not a species uses core or edge habitat (with a patch size/abundance relationship more likely to occur for core habitat species). This paper would be much better served by clarifying this context from the beginning, in the introduction, and then motivating why this relationship might require a new analysis for the case of riverine fish (e.g., could advection change these expectations?).

Second, the description of the potential applications of the findings here to biodiversity conservation (abstract lines 43-46, discussion lines 385-396) could use greater specificity: how exactly might one use these findings to prioritize sites for habitat restoration? Also, wouldn't dam removal (line 393) lead to more lotic "matrix" habitats rather than lentic "patch" habitats, such that there'd be a trade-off with dam removal between positive effects on connectivity and negative effects on patch number and size for lentic species (as indicated on lines 80-81)?

Third, for the results on population variability, to claim support for the portfolio effect specifically (lines 350-353), the authors must demonstrate that the coefficient of variation would be greater if all sub-populations were a single homogenous population instead of divided into sub-populations (see the analysis in the Schindler et al 2010 paper cited here). The portfolio effect means not just lower variability, but specifically lower variability of a meta-population arising from independence of sub-populations (e.g., each sub-population having independent sub-population-level fluctuations due to factors such as localized environmental variability or local adaptation, so a bad year for one sub-population isn't necessarily a bad year for all sub-populations). In addition, while larger population sizes in sub-populations can increase portfolio-level buffering, anything that increases connectivity can, in some cases, reduce portfolio buffering by increasing cross-sub-population synchrony. See:

Abbott, K. C. (2011). A dispersal-induced paradox: synchrony and stability in stochastic metapopulations. *Ecology Letters*, 14(11), 1158-1169.

Bjørnstad, O. N., Ims, R. A., & Lambin, X. (1999). Spatial population dynamics: analyzing patterns and processes of population synchrony. *Trends in Ecology & Evolution*, 14(11), 427-432.

Liebhold, A., Koenig, W. D., & Bjørnstad, O. N. (2004). Spatial synchrony in population dynamics. *Annu. Rev. Ecol. Evol. Syst.*, 35, 467-490.

for more on this. If, instead of doing a portfolio effect analysis to support their claim here in the discussion, the authors want to connect their results to the portfolio effect concept without additional analyses, then they would need to recognize its multiple facets (e.g., the potential role of connectivity in synchronizing populations) in a more careful and nuanced discussion and be careful to avoid wording that suggests support for the portfolio effects from their results.

Additional comments:

Line 89: does exploitation modify habitat as implied here? In marine systems, that depends on the fishing gear (with bottom trawling and dynamite fishing in particular causing habitat damage), but I'm not aware of how this occurs in freshwater habitats.

Line 97: why is this a "sad" outcome? Delete "sadly" or use a less normative term (although I expect this sentence will change more substantively in response to the comments above).

Line 143 "that we regarded highly": this phrasing is a bit awkward, maybe "The fact that spatial data like this can be created entirely by computer is valuable because...".

Line 179-180 "the next by": not sure what this phrase is referring to, perhaps delete?

Line 194: I'm guess there were no data on habitat quality and fishing pressure, but if there were, that'd be quite useful to include in the analyses.

Line 280: for improved clarity here, specify "previous studies" as "occurrence studies of riverine fish", and then replace "As such" with "For example"

Line 315: is a 0.7% difference in AIC really "significant"? It's an improved fit, but seems like a small one.

Lines 369-372: please indicate percent of variation explained to back this up

Lines 373-4 "promote/increase/stabilize/reduce": this phrasing suggests causality was demonstrated, while the data here are observational; either change to phrasing like "are associated with" or soften with "likely" or "might".

Referee: 2

Comments to the Author(s)

In this study, authors exploited a large Swedish electrofishing database to assess the roles of lotic habitat size and isolation in shaping the occurrence, density and stability of brown trout subpopulations.

This is a very interesting and robust study, with striking results that match theoretical expectations surprisingly well (given the use of empirical data). It is rather well written and nice to read. I have two main comments though.

My first comment is about statistical analyses. I think that models should have been run with two (nested) random effects, while the authors only considered a single random effect. This would probably not alter the results and interpretations (that seem pretty robust), but this is to be checked.

My second comment is about the discussion, which gets poorer toward the end. The two last sections seem rather incongruous. The former (about between age classes comparisons) could be disseminated throughout other sections, while the latter (about so-called large-scale patterns) is not supported by any clearly detailed hypotheses or statistical analysis and thus appears highly speculative (if not flawed). I suggest that the authors simply delete this last section, or provide full details.

See detailed comments below.

SUMMARY

No specific comment

INTRODUCTION

L.100-103: "autecological mechanisms": This term, as well as related mentioned mechanisms (immigration behaviours and edge effects) are detailed in Bowman et al. 2002 (cited reference) but would here require additional information for readers to fully grasp your statement. This is all the more important that these mechanisms are central in your study.

MATERIAL AND METHODS

L.126: "fastighetskartan": this must be a Swedish word for some kind of map, but please, provide a English translation.

L.134: "with all stretches": with all other stretches?

L.183: I think that this section is more about "model choice" or "model specification" than about "model selection" (the latter referring to the specific statistical approach of selecting the most parsimonious model among a set of candidate models)

L.195-196: Your design is actually nested: several data points within each lotic stretch, and several stretches within each river. With a single random effect (lotic stretch), you take into consideration the "dependency between data points" but you do not take into consideration the fact that stretches from a same river might provide higher/lower yields. I think that you need two nested random effects: a "stretch" random effect, nested into a "river" random effect. This might not change your results drastically, but I think it would be preferable... all the more so as you actually identified an effect of rivers on densities (Table 3... but see my comment about L.360-366)

L.203 and L.227: ANOVAs with non-linear trends are not reported in the manuscript (unless I am mistaken, only results of ANOVAs with linear trends are reported, in L. 254 and L. 336). Is there a reason for that? Were there inconclusive? Furthermore, could you provide the R-package used for these analyses?

L.229: "No random effects were applied.": could you explain why? as for densities, I guess that CV could be influenced by both site- and river-peculiarities.

RESULTS & DISCUSSION

L.297: Bender et al.

L.315: please, specify which model (1 or 2) is associated with each AIC

L.320: or even a combination of both? (e.g. 0+ mortality and 1+ behavioral avoidance)

L.355-358: this section seems a bit disconnected from the rest of the discussion and could be merged with the corresponding sections...

L.360-366: this section, as the previous one, is rather “out of the blue”, and quite obscure, since the corresponding analytical approach is not provided in the method section (unless I missed something). I cannot figure out what you actually did here... Was it a ZINB mixed model? A subset of Model 1?... From my understanding, eta-squared are computed when the model includes categorical predictors, but I think that stretch length and connectivity are continuous predictors here... are you sure of the use of eta-squared? Apparently, you used rivers as a random effect, since discarding the river effect makes you switch from a GLMM to a GLM (caption in Table 3; see by the way my comment about L.195), but at the same time, you mention in L.364 that you used a “random effect of habitat identity” (and not “river identity”): this is to be clarified. Finally, why did you run the model with and without the random effect, and how can you draw any (or such) biological conclusions thereby? All of this has to be clarified... or the whole section just discarded, since the remaining of the manuscript is quite convincing and this section does not add any relevant/robust interpretations.

CONCLUSION

No specific comment

FIGURES AND TABLES

Table 1: “near” versus “close” : you should use a single term

Figure 2 and Table 2: the former uses the term “directionality” (which makes sense and is also used in the main text in L.215) whereas the latter uses the term “upstream”. You should use “directionality” throughout the text (with for instance “Directionality (U)” in Table 2)

Table 3: the statistical approach corresponding to these outputs cannot be found anywhere (or at least, not in a straightforward way). See my comments about L. 360-366.

Author's Response to Decision Letter for (RSPB-2021-1255.R0)

See Appendix A.

RSPB-2021-1255.R1 (Revision)

Review form: Reviewer 1

Recommendation

Accept with minor revision (please list in comments)

Scientific importance: Is the manuscript an original and important contribution to its field?

Good

General interest: Is the paper of sufficient general interest?

Good

Quality of the paper: Is the overall quality of the paper suitable?

Good

Is the length of the paper justified?

Yes

Should the paper be seen by a specialist statistical reviewer?

No

Do you have any concerns about statistical analyses in this paper? If so, please specify them explicitly in your report.

No

It is a condition of publication that authors make their supporting data, code and materials available - either as supplementary material or hosted in an external repository. Please rate, if applicable, the supporting data on the following criteria.

Is it accessible?

Yes

Is it clear?

Yes

Is it adequate?

Yes

Do you have any ethical concerns with this paper?

No

Comments to the Author

The revised version mostly addresses my previous comments, with a few lingering minor points:

Thank you for moving the context-dependency of use of core or edge habitat determining the habitat size-abundance relationship to the Introduction. However, splitting this information between the Intro paragraph 3 and the second half of paragraph 4 is a bit disjointed; it would read better to have it in one place (e.g., first make the point about occupancy vs. abundance relationships [first half of paragraph 4], then get to what is known about abundance relationships [paragraph 3] including their context-dependency [second half of paragraph 4]).

In addition, it'd still be good to make the (now-deleted) point in the Discussion that the results here support the context-dependency of a positive abundance-habitat size relationship for core habitat users given lotic habitat as core habitat for trout; the lead in to that point could just be much shorter than previously given that this context is now established in the Introduction.

The text on portfolio effects now better recognizes the potential synchronizing role of connectivity, but I still don't agree that the results here "indicate that increased connectivity can lead to dispersal-induced stability that goes beyond that", as the results here are about within-population variability, not across-population variability. Therefore, please specify that "lotic stretches can have a stabilizing effect on individual populations" in the second-to-last sentence of this paragraph, and a better concluding point (instead of the text quoted above) would be to then say that a remaining unknown is (or next step would be to see) how the individual-population-level stabilizing effect balances with the potential synchronizing effect of increased diversity to determine the overall effect on meta-population level variability.

The Discussion text on management implications is more specific, but the last sentence of the abstract on this point is still a bit vague ("stressing the importance of habitat size and connectivity" for what? "reduce negative impacts" how?). Using much of the same language but

more directly, I'd suggest something like "These results can inform prioritization of sites for habitat restoration, dam removal and reintroduction by highlighting the role of suitable habitat size and connectivity in population abundance and stability for riverine fish populations."

In addition, in the new Discussion text, please avoid prescriptive language ("one should prioritize measure types and locations...") in recognition that it is the role of the scientist to determine the most effective way to achieve a given goal while it is the role of managers and stakeholders to determine if that is a relevant goal (i.e., what "should" be). Therefore, please re-frame this text in terms of what goal or outcome prioritizing such sites can more effectively achieve.

Decision letter (RSPB-2021-1255.R1)

27-Sep-2021

Dear Mr Tamario

I am pleased to inform you that your Review manuscript RSPB-2021-1255.R1 entitled "Size, connectivity and edge effects of stream habitats explain spatiotemporal variation in brown trout (*Salmo trutta*) density" has been accepted for publication in Proceedings B.

The referee(s) do not recommend any further changes. Therefore, please proof-read your manuscript carefully and upload your final files for publication. Because the schedule for publication is very tight, it is a condition of publication that you submit the revised version of your manuscript within 7 days. If you do not think you will be able to meet this date please let me know immediately.

To upload your manuscript, log into <http://mc.manuscriptcentral.com/prsb> and enter your Author Centre, where you will find your manuscript title listed under "Manuscripts with Decisions." Under "Actions," click on "Create a Revision." Your manuscript number has been appended to denote a revision.

You will be unable to make your revisions on the originally submitted version of the manuscript. Instead, upload a new version through your Author Centre.

- 1) A text file of the manuscript (doc, txt, rtf or tex), including the references, tables (including captions) and figure captions. Please remove any tracked changes from the text before submission. PDF files are not an accepted format for the "Main Document".
- 2) A separate electronic file of each figure (tiff, EPS or print-quality PDF preferred). The format should be produced directly from original creation package, or original software format. Please note that PowerPoint files are not accepted.
- 3) Electronic supplementary material: this should be contained in a separate file from the main text and the file name should contain the author's name and journal name, e.g `authorname_procb_ESM_figures.pdf`

All supplementary materials accompanying an accepted article will be treated as in their final form. They will be published alongside the paper on the journal website and posted on the online figshare repository. Files on figshare will be made available approximately one week before the accompanying article so that the supplementary material can be attributed a unique DOI. Please see: <https://royalsociety.org/journals/authors/author-guidelines/>

4) Data-Sharing and data citation

It is a condition of publication that data supporting your paper are made available. Data should be made available either in the electronic supplementary material or through an appropriate repository. Details of how to access data should be included in your paper. Please see <https://royalsociety.org/journals/ethics-policies/data-sharing-mining/> for more details.

<http://datadryad.org/submit?journalID=RSPB&manu=RSPB-2021-1255.R1> which will take you to your unique entry in the Dryad repository.

Once again, thank you for submitting your manuscript to Proceedings B and I look forward to receiving your final version. If you have any questions at all, please do not hesitate to get in touch.

Sincerely,

Dr Daniel Costa

Associate Editor Board Member: 1

Comments to Author:

The revisions have improved the paper. There remain a few minor but specific edits to the text that the referee has suggested and which would help to further clarify the paper.

Reviewer(s)' Comments to Author:

Referee: 1

Comments to the Author(s)

The revised version mostly addresses my previous comments, with a few lingering minor points:

Thank you for moving the context-dependency of use of core or edge habitat determining the habitat size-abundance relationship to the Introduction. However, splitting this information between the Intro paragraph 3 and the second half of paragraph 4 is a bit disjointed; it would read better to have it in one place (e.g., first make the point about occupancy vs. abundance relationships [first half of paragraph 4], then get to what is known about abundance relationships [paragraph 3] including their context-dependency [second half of paragraph 4]).

In addition, it'd still be good to make the (now-deleted) point in the Discussion that the results here support the context-dependency of a positive abundance-habitat size relationship for core habitat users given lotic habitat as core habitat for trout; the lead in to that point could just be much shorter than previously given that this context is now established in the Introduction.

The text on portfolio effects now better recognizes the potential synchronizing role of connectivity, but I still don't agree that the results here "indicate that increased connectivity can lead to dispersal-induced stability that goes beyond that", as the results here are about within-population variability, not across-population variability. Therefore, please specify that "lotic stretches can have a stabilizing effect on individual populations" in the second-to-last sentence of this paragraph, and a better concluding point (instead of the text quoted above) would be to then say that a remaining unknown is (or next step would be to see) how the individual-population-level stabilizing effect balances with the potential synchronizing effect of increased diversity to determine the overall effect on meta-population level variability.

The Discussion text on management implications is more specific, but the last sentence of the abstract on this point is still a bit vague ("stressing the importance of habitat size and connectivity" for what? "reduce negative impacts" how?). Using much of the same language but more directly, I'd suggest something like "These results can inform prioritization of sites for habitat restoration, dam removal and reintroduction by highlighting the role of suitable habitat size and connectivity in population abundance and stability for riverine fish populations." In addition, in the new Discussion text, please avoid prescriptive language ("one should prioritize measure types and locations...") in recognition that it is the role of the scientist to determine the most effective way to achieve a given goal while it is the role of managers and stakeholders to determine if that is a relevant goal (i.e., what "should" be). Therefore, please re-frame this text in terms of what goal or outcome prioritizing such sites can more effectively achieve.

Decision letter (RSPB-2021-1255.R2)

28-Sep-2021

Dear Mr Tamario

I am pleased to inform you that your manuscript entitled "Size, connectivity and edge effects of stream habitats explain spatiotemporal variation in brown trout (*Salmo trutta*) density" has been accepted for publication in Proceedings B.

Data Accessibility section

Open Access

Paper charges

Sincerely,
Proceedings B
mailto:proceedingsb@royalsociety.org

Appendix A

** The authors' responses to reviewers' comments are in dark red and pre-faced by two asterisks.

Referee: 1

Comments to the Author(s)

This paper provides an analysis to show that patch size and distance correlate with population density. This is a fairly fundamental question in spatial ecology, well-explored in a variety of taxa as the authors reference, while this paper investigates it in the context of riverine fish in particular. My first main feedback is one of this framing: currently, the introductory text on the relationship between patch size and population sizes in other taxa (lines 93-99) might be taken to imply that there is no support for such a relationship. It is not until the discussion (lines 291-298) that the authors clarify that half of studies in the referenced meta-analyses do support such a relationship, and a common context-dependency for why mixed results might occur is whether or not a species uses core or edge habitat (with a patch size/abundance relationship more likely to occur for core habitat species). This paper would be much better served by clarifying this context from the beginning, in the introduction, and then motivating why this relationship might require a new analysis for the case of riverine fish (e.g., could advection change these expectations?).

** We thank the reviewer for the useful comments.

** We have moved the concerning lines (291-298) from the discussion to the introduction and further developed the reasoning for the need to consider autecological mechanisms (i.e. the context-dependency) as per another comment. We believe the introduction reads better now, thanks to this.

Second, the description of the potential applications of the findings here to biodiversity conservation (abstract lines 43-46, discussion lines 385-396) could use greater specificity: how exactly might one use these findings to prioritize sites for habitat restoration? Also, wouldn't dam removal (line 393) lead to more lotic "matrix" habitats rather than lentic "patch" habitats, such that there'd be a trade-off with dam removal between positive effects on connectivity and negative effects on patch number and size for lentic species (as indicated on lines 80-81)?

** We have added statements to the revised Discussion section that clarify how one might apply our findings (for details please see the Track Changes-document): "Based on the results here, one should prioritize measure types and locations that enable previously isolated stretches to be connected, and the larger the stretches the higher expected density and stability of populations. Dam removals, for example, could restore connection but also create new lotic habitats from previously inundated areas, generating both productivity in the restored lotic stretch *and* new migration stepping stones for rheophilic fish."

** Regarding the reviewer's second comment concerning line 393, there seems to be a misunderstanding. As the reviewer writes, dam removal would lead to more lotic habitats, but lotic stretches are not "matrix" habitats but "patch" habitats, so dam removal does indeed increase "patch" habitats.

Third, for the results on population variability, to claim support for the portfolio effect specifically (lines 350-353), the authors must demonstrate that the coefficient of variation would be greater if all sub-populations were a single homogenous population instead of divided into sub-populations (see the analysis in the Schindler et al 2010 paper cited here). The portfolio effect means not just lower variability, but specifically lower variability of a meta-population arising from independence of sub-populations (e.g., each sub-population having independent sub-population-level fluctuations due to factors such as localized environmental variability or local adaptation, so a bad year for one sub-population isn't necessarily a bad year for all sub-populations). In addition, while larger population sizes in sub-populations can increase portfolio-level buffering, anything that increases connectivity can, in some cases, reduce portfolio buffering by increasing cross-sub-population synchrony.

See:

Abbott, K. C. (2011). A dispersal- induced paradox: synchrony and stability in stochastic metapopulations. *Ecology Letters*, 14(11), 1158-1169.

Bjørnstad, O. N., Ims, R. A., & Lambin, X. (1999). Spatial population dynamics: analyzing patterns and processes of population synchrony. *Trends in Ecology & Evolution*, 14(11), 427-432.

Liebold, A., Koenig, W. D., & Bjørnstad, O. N. (2004). Spatial synchrony in population dynamics. *Annu. Rev. Ecol. Evol. Syst.*, 35, 467-490.

for more on this. If, instead of doing a portfolio effect analysis to support their claim here in the discussion, the authors want to connect their results to the portfolio effect concept without additional analyses, then they would need to recognize its multiple facets (e.g., the potential role of connectivity in synchronizing populations) in a more careful and nuanced discussion and be careful to avoid wording that suggests support for the portfolio effects from their results.

**** This is a very insightful point and we agree with this. We have reordered this paragraph and added a sentence to convey the multiple facets of the portfolio effect (for details see the Track Changes document).**

Additional comments:

Line 89: does exploitation modify habitat as implied here? In marine systems, that depends on the fishing gear (with bottom trawling and dynamite fishing in particular causing habitat damage), but I'm not aware of how this occurs in freshwater habitats.

**** Exploitation means “the action of making use of and benefitting from resources”. In this case, we meant the river as a resource and the subject of exploitation (thus by damming, channeling, clearing, and other habitat modifying activities) rather than exploitation by fishing. The latter, as the reviewer suggests, does not alter the habitat that much. To clarify and avoid further misunderstandings, we have modified the sentence such that it now reads “A better understanding of what determines the distribution and abundance of riverine fish, and informed projections of how they might respond to management actions and exploitations (e.g., damming, channeling and clearings) that modify river habitats, is therefore in great demand (26). “**

Line 97: why is this a "sad" outcome? Delete "sadly" or use a less normative term (although I expect this sentence will change more substantively in response to the comments above).

** Fully agreed. We removed “Sadly, “

Line 143 "that we regarded highly": this phrasing is a bit awkward, maybe "The fact that spatial data like this can be created entirely by computer is valuable because..."?

** We changed the statement to “That spatial data like this can be created entirely by computer is valuable because of the possibility to extrapolate to larger contexts.” as we already touched upon individuals error corrections in the previous sentence.

Line 179-180 "the next by": not sure what this phrase is referring to, perhaps delete?

** We agree that this was awkwardly phrased. We replaced “next by GIS” with “closest” and added “neighboring”. The sentence now reads “Instead, we used the closest predicted neighbouring habitat (and their size), regardless of whether it was occupied or not, as a proxy of connectivity. “

Line 194: I'm guess there were no data on habitat quality and fishing pressure, but if there were, that'd be quite useful to include in the analyses.

** The electrofishing efforts provide some data on habitat characteristics, but we preferred to keep it simpler by letting the random effects of “lotic stretch” mainly take care of that by treating it as random among-site variation. There are many other articles that focus more directly and extensively on habitat requirements/preferences of brown trout (e.g., Armstrong et al. (2003) Habitat requirements of Atlantic salmon and brown trout in rivers and streams. Fisheries Research). Widespread data on fishing pressure does not exist in this area.

Line 280: for improved clarity here, specify "previous studies" as "occurrence studies of riverine fish", and then replace "As such" with "For example"

** Thank you for the suggestion, we changed it accordingly.

Line 315: is a 0.7% difference in AIC really "significant"? It's an improved fit, but seems like a small one.

** We agree that raw AIC values can be a bit difficult to interpret unambiguously. To ease interpretation, we converted the AIC values to Akaike weights, which instead represent the relative likelihood of a model. The Akaike weights shows 99.1% favor for model 2 (AIC = 1828) and 0.9% favor for model 1 (AIC = 1837). The fact that model 2 produces reduced AIC when adding three new terms indicates that the variables provide more new information to the model (not present in the already included variables) than what is penalized for adding them.

Lines 369-372: please indicate percent of variation explained to back this up

** We added marginal R2 values as received from model 1 & 2.

Lines 373-4 "promote/increase/stabilize/reduce": this phrasing suggests causality was demonstrated, while the data here are observational; either change to phrasing like "are associated with" or soften with "likely" or "might".

** Agreed. The sentence now reads: “The analyses provide evidence **suggesting that** large and well connected rapid-flowing riverine habitats positively promote habitat occupancy, increase and stabilize population density, and reduce local extinction rates of brown trout - *S. trutta*.”

Referee: 2

Comments to the Author(s)

In this study, authors exploited a large Swedish electrofishing database to assess the roles of lotic habitat size and isolation in shaping the occurrence, density and stability of brown trout subpopulations.

This is a very interesting and robust study, with striking results that match theoretical expectations surprisingly well (given the use of empirical data). It is rather well written and nice to read. I have two main comments though.

** We thank the reviewer for the appreciative and useful comments.

My first comment is about statistical analyses. I think that models should have been run with two (nested) random effects, while the authors only considered a single random effect. This would probably not alter the results and interpretations (that seem pretty robust), but this is to be checked.

** We agree with the reviewer and we have re-analyzed our zero-inflated negative binomial models with **two** levels of random effects, river identity and lotic habitat identity, using R package glmmTMB. The conclusions drawn from the results are unaffected by re-analyzing but because of some minor dissimilarities between models we reordered the result section slightly.

My second comment is about the discussion, which gets poorer toward the end. The two last sections seem rather incongruous. The former (about between age classes comparisons) could be disseminated throughout other sections, while the latter (about so-called large-scale patterns) is not supported by any clearly detailed hypotheses or statistical analysis and thus appears highly speculative (if not flawed). I suggest that the authors simply delete this last section, or provide full details.

** Please note that we state in the Methods section that the results were qualitatively similar when age classes were analyzed together and separately. We therefore did not see the need to repeat this similarity in/after every result section. We have now deleted the “Comparisons between age classes” section. For those that are interested, we still reference to both figures and tables in the Supporting Information at the appropriate places throughout the results.

** We deleted the “Large-scale patterns” section, as suggested by the reviewer.

See detailed comments below.

SUMMARY

No specific comment

INTRODUCTION

L.100-103: “autecological mechanisms”: This term, as well as related mentioned mechanisms (immigration behaviours and edge effects) are detailed in Bowman et al. 2002 (cited reference) but would here require additional information for readers to fully grasp your statement. This is all the more important that these mechanisms are central in your study.

** Thank you for this suggestion. We have clarified this so that this part of the paragraph now reads: “That reviews and meta-analyses suggest that there is no consistent relationship between patch size and population density (30-34) emphasizes the importance of considering autecological mechanisms, i.e. how specific species interact with their environment. Migration behaviour, and the effects of habitat edges and interactions with other species may affect the direction of the predictions (33). For example, Bender et al. (31) report that patch size effects were negative for species that use edge habitats and positive for species that use core habitat.”

MATERIAL AND METHODS

L.126: “fastighetskartan”: this must be a Swedish word for some kind of map, but please, provide a English translation.

** “Fastighetskartan” translates to “the property map” and is referred to as such by Lantmäteriet (the Swedish mapping, cadastral and land registration authority). We have clarified this in the manuscript. The sentence now reads “This was accomplished by combining vector lines of rivers as obtained from the Property Map (fastighetskartan, 1:10 000) and altitude data from the national height data model (GSD-Höjddata grid 2+), the highest resolution layer available of each (acquired from Lantmäteriet; the Swedish mapping, cadastral and land registration authority), respectively, in Sweden.”

L.134: “with all stretches”: with all other stretches?

** The sentence now reads: “A total of 714 lotic stretches were identified with all other river stretches in-between considered lentic matrix habitat (**Fig. 1c** and **Fig S2**).”

L.183: I think that this section is more about “model choice” or “model specification” than about “model selection” (the latter referring to the specific statistical approach of selecting the most parsimonious model among a set of candidate models)

** Agreed. We have changed the header to now read “*Data handling and model choice*”.

L.195-196: Your design is actually nested: several data points within each lotic stretch, and several stretches within each river. With a single random effect (lotic stretch), you take into consideration the “dependency between data points” but you do not take into consideration the fact that stretches from a same river might provide higher/lower yields. I think that you need two nested random effects: a “stretch” random effect, nested into a “river” random effect. This might not change your results drastically, but I think it would be preferable... all the more so as you actually identified an effect of rivers on densities (Table 3... but see my comment about L.360-366)

** We agree and we have solved this issue, see above and the revised manuscript for details.

L.203 and L.227: ANOVAs with non-linear trends are not reported in the manuscript (unless I am mistaken, only results of ANOVAs with linear trends are reported, in L. 254 and L. 336). Is there a reason for that? Were there inconclusive? Furthermore, could you provide the R-package used for these analyses?

** Because the quadratic and cubic contrasts were not significant in any of the tests we did not include those results in the first version. We have now added them to the revised version. The analyses were performed in “stats”-package in R following the advice of a colleague. Find the R-code below:

```
df$IBGpoly <- factor(df$IBG, levels=c("A","B","C","D"))

contrasts(df$IBGpoly) <- contr.poly

round(crossprod(contrasts(df$IBGpoly)),2)

aov_out <- aov(density ~ IBGpoly, df)

summary(aov_out, split = list(IBGpoly=list("linear" = 1, "quadratic" = 2, "cubic" = 3)))
```

L.229: “No random effects were applied.”: could you explain why? as for densities, I guess that CV could be influenced by both site- and river-peculiarities.

** One reason for considering random effects of rivers and stretches when evaluating the associations of trout density with habitat size and habitat isolation was that any differences in average densities may obscure such associations. However, this is less of a problem when analyzing density fluctuations, as these were quantified using CV – which at least in theory should be independent of the mean densities. Based on our understanding, including random effects in the analyses of CV would answer the question whether there are any effects on density fluctuations of river or lotic stretch *beyond* those mediated by habitat size and connectivity – and answering that question it outside the main scope of this contribution.

RESULTS & DISCUSSION

L.297: Bender et al.

** Thank you, fixed.

L.315: please, specify which model (1 or 2) is associated with each AIC

** Thank you, fixed.

L.320: or even a combination of both? (e.g. 0+ mortality and 1+ behavioral avoidance)

** Of course, we added this to the sentence.

L.355-358: this section seems a bit disconnected from the rest of the discussion and could be merged with the corresponding sections...

**** This section is now removed and we added a small statement on the similarities of effects for both age classes in the Discussion.**

L.360-366: this section, as the previous one, is rather “out of the blue”, and quite obscure, since the corresponding analytical approach is not provided in the method section (unless I missed something). I cannot figure out what you actually did here... Was it a ZINB mixed model? A subset of Model 1?... From my understanding, eta-squared are computed when the model includes categorical predictors, but I think that stretch length and connectivity are continuous predictors here... are you sure of the use of eta-squared? Apparently, you used rivers as a random effect, since discarding the river effect makes you switch from a GLMM to a GLM (caption in Table 3; see by the way my comment about L.195), but at the same time, you mention in L.364 that you used a “random effect of habitat identity” (and not “river identity”): this is to be clarified. Finally, why did you run the model with and without the random effect, and how can you draw any (or such) biological conclusions thereby? All of this has to be clarified... or the whole section just discarded, since the remaining of the manuscript is quite convincing and this section does not add any relevant/robust interpretations.

**** We agree that this section, and the associated tests, do not add anything major to the main scope of the manuscript so we decided to remove it.**

CONCLUSION

No specific comment

FIGURES AND TABLES

Table 1: “near” versus “close” : you should use a single term

**** We changed this so that it states “close” on both locations.**

Figure 2 and Table 2: the former uses the term “directionality” (which makes sense and is also used in the main text in L.215) whereas the latter uses the term “upstream”. You should use “directionality” throughout the text (with for instance “Directionality (U)” in Table 2)

**** We agree and changed the table according to the suggestions.**

Table 3: the statistical approach corresponding to these outputs cannot be found anywhere (or at least, not in a straightforward way). See my comments about L. 360-366.

**** We removed this test so this comment does not apply any more.**